# Customer Involvement in Sustainable Tourism Planning at Lake Balaton, Hungary—Analysis of the Consumer Preferences of the Active Cycling Tourists

**Katalin Lőrincz [1], Zsuzsanna Banász [2] and János Csapó [3,\*]**

[1] Department of Tourism, University of Pannonia, Egyetem str. 10, H-8200 Veszprém, Hungary; lorincz.katalin@gtk.uni-pannon.hu

[2] Department of Quantitative Methods, University of Pannonia, Egyetem str. 10, H-8200 Veszprém, Hungary; banasz.zsuzsanna@gtk.uni-pannon.hu

[3] Institute of Marketing and Tourism, Faculty of Business and Economics, University of Pécs, Rákóczi str. 80, H-7622 Pécs, Hungary

\* Correspondence: csapo.janos@ktk.pte.hu

**Abstract:** This study uses an innovative tourism product development approach, based on co-creation or customer involvement, related to Lake Balaton, a mass tourism-based destination in Hungary, from the point of view of the market segment of active cycling tourists. The investigation of opportunities for the development of cycling tourism first of all relies on the new approach of attraction and product development around the destination, in which it is important to take into consideration the consumer preferences of the most important related group of tourists—active cycling tourists. The sustainable approach of tourism product development also provides an opportunity to decrease the spatial and temporal concentration of tourism, which is largely concentrated on the summertime season. The aim of this study is to explore aspects of the customers' demand for tourism development in terms of cycling tourism with the help of primary data collection, in order to provide adequate directions for sustainable tourism development in the destination. Revealing the demand side of active cycling tourism related to Lake Balaton, the authors used both qualitative (focus group discussions and structured interviews) and quantitative questionnaire survey (computer-assisted data collection) research methods. The latter online surveys were carried out in November and December, 2019, and resulted with an appraisable sample of 809 questionnaires. As for the method, descriptive statistics and relationship analyses were applied. More than five thousand (5050) possible relationships were examined between the closed answers of the questionnaire by Kendall's rank correlation coefficient ($\tau$) and Cramer's V, depending on whether they could be measured on a nominal or ordinal scale. The results show that the content analysis of the primary research provides well determined directions for the sustainable tourism development of cycling tourism at Lake Balaton, so customer involvement seems to be a win-win situation both for the customers (tourists) and the decision makers.

**Keywords:** sustainable tourism development; Lake Balaton; cycling tourism; consumer preferences

## 1. Introduction

The Balaton destination is a traditionally prominent tourism region of Hungary, playing a prioritized and significant role in the tourism industry of the country [1]. For domestic tourists, it is the most well-known and visited holiday region of the country, and for foreign visitors, the lake and its surrounding region is the second most popular destination [2]. Based on its unique natural and cultural endowments, it has been one of the most popular tourism destinations from a historical perspective as well [3–5]. In the 2010s, besides passive waterside tourism activities, auxiliary tourism

products such as cultural and festival tourism, health tourism and active tourism (of which cycling tourism has become the most dominant product) have been strengthened [3]. The strengthening of these auxiliary products has increased the spatial extent of tourism-involved regions, and due to the developments of the last decades, besides the traditional ones, new central regions have appeared in the Balaton region [6–8].

The Balaton Accentuated Tourism Development Region, impounded by a government decree [9], is the most popular destination in the country among domestic visitors and the second most popular among foreign visitors. The region covers 174 settlements, of which 42 are located directly along the shore [10]. The region should not be confused with the Balaton Accentuated Tourism Zone (with 179 settlements). Although the overlap between these two regions is dominant, there are differences, for instance, in this case, a county seat (Veszprém) is also involved (Figure 1) [11].

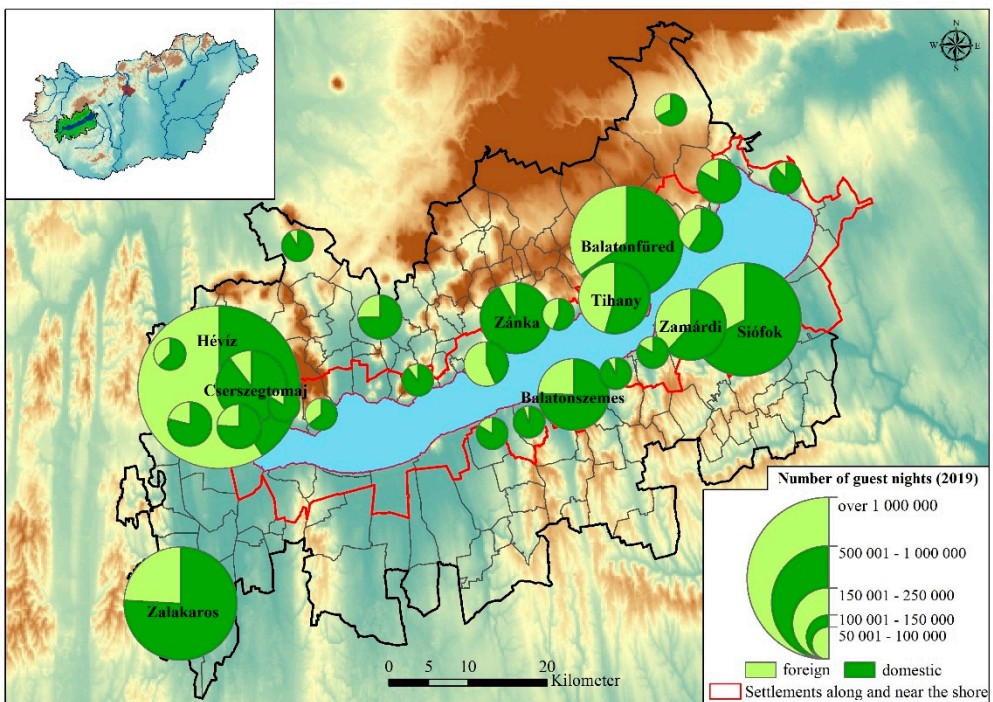

**Figure 1.** The Balaton Accentuated Tourism Zone and its most important settlements (2019). Source: the authors.

Lake Balaton, the greatest freshwater lake in Central Europe, attracts visitors with a supply oriented to seasonal changes [3,12,13]. Of course, the most important natural attraction is the lake itself where water and waterside tourism are the most important elements of attraction (boating, sailing, bathing, water sports, kayaking, canoeing, etc.). There are more than 100 beaches along the shore, qualified by the Blue Flag qualification system. The water depth on the northern shore is deeper, while on the southern shore it is much shallower, providing different physical endowments for different target groups (e.g., families with small children prefer the southern shore).

The natural attractions of the lake are complemented with such regions and attractions as the Balaton Highlands National Park (hiking and trekking), the Hévíz curative lake (health tourism, recreation) and the Balaton Cycle Circuit (cycling, jogging, roller skating).

The total length of the Balaton Cycle Circuit is 205 km. The route basically follows the shore of the lake and consists of designated cycle routes, as well as public roads and byways. The signs are not consistently established and the quality, and thus the level of safety, of even the paved roads is problematic. The route already contains loops or sub-itineraries that are not yet integrated well into the cycle circuit (Figure 2).

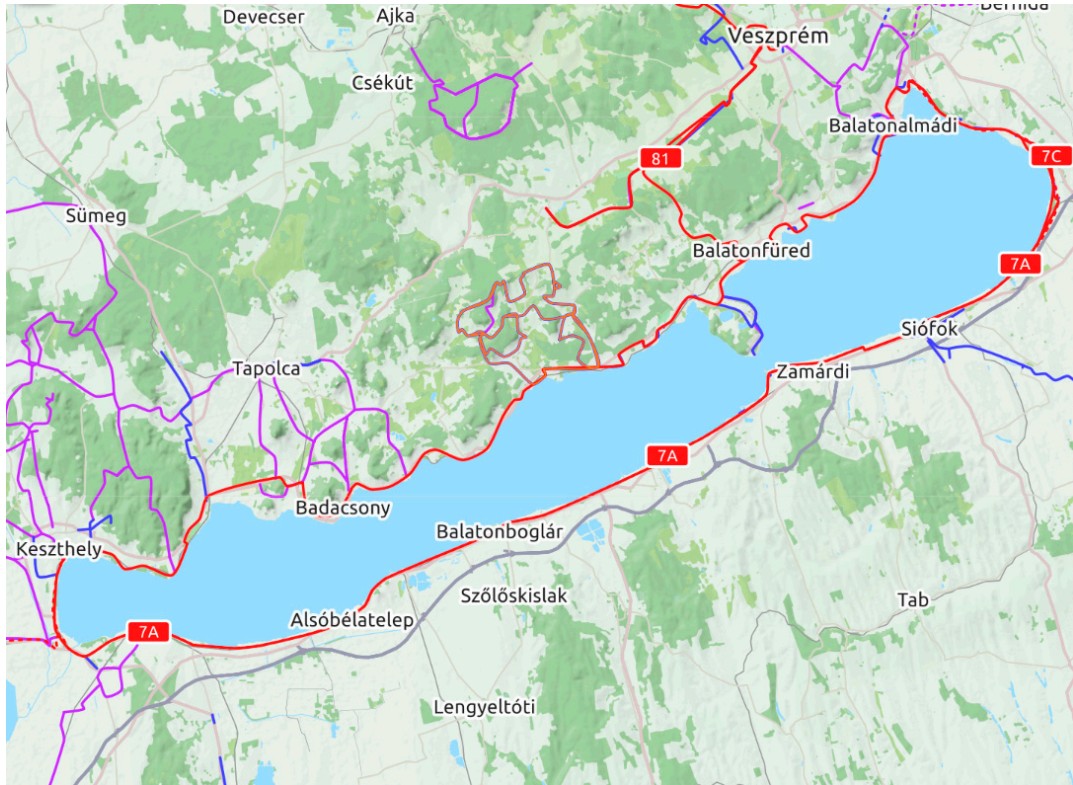

**Figure 2.** The Balaton Cycle Circuit and the designated cycling routes around Lake Balaton. Source: https://www.futas.net/terkep/magyarorszag/kerekparut-terkep.php.

According to the latest demand statistics, until 2019 the tourism flow of Lake Balaton showed an increasing trend. In 2019, the destination attracted 16% of all the guests in the country and 20% of all the guest nights (Figure 3). It should also be highlighted that 77% of the guests in the destination were domestic tourists, with 68% of all the guest nights (Figure 4). The mean length of stay was 3.07 days (4.27 days for foreign tourists, 2.71 days for domestic tourists) [14].

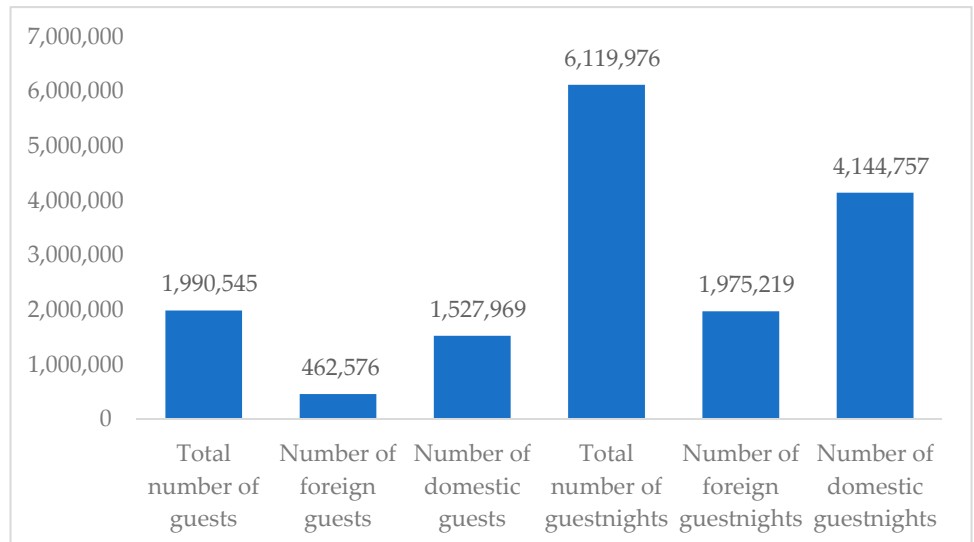

**Figure 3.** The tourism flow of the Balaton destination in commercial accommodations (2019). Source: Based on HSCO (Hungarian Central Statistical Office) database [14], own editing.

In terms of the foreign market, the most important tourist-sending regions are Germany (97,000 tourists), the Czech Republic (57,000 tourists), Austria (56,900 tourists) and Russia (25,600 tourists). Based on the guest nights, Germany (609,000 guest nights) is followed by Russia (215,000 guest nights), the Czech Republic (207,000 guest nights) and Austria (197,000 guest nights).

The level of seasonality is still high in the Balaton region; however, recent trends demonstrated that the supply in the spring and autumn periods can be diversified and so the measure of seasonality can be decreased. Based on this product development strategy, active tourism (and within that, cycling tourism) plays a highlighted role in the region.

The present study intends to demonstrate the primary research results of the authors, which were carried out in order to survey and reveal real market demand for the sustainable tourism development of Lake Balaton, Hungary. This kind of approach in tourism development is not yet widely applied in the Hungarian tourism industry, which is still very much characterized by top–bottom initiatives. The authors believe that, based on international examples where the involvement of the market players has a more highlighted role in tourism planning, this kind of approach can strengthen the role of the other end of the planning process, namely, the customers. Moreover, the present paper can support the traditional destination-based approach with a new, attraction-, service- and basic infrastructure-based development approximation, relying on the needs of the real demand. In order to do this, the authors surveyed the demand of the potential active cycling tourists of Lake Balaton, Hungary.

The authors agree that this approach can assure the economic, social and environmental sustainability of the developed attractions parallel with the primary objective of the developments, namely, that in the future a growing number of quality experiences should be provided for the tourists of the destination, combining tourist's needs with destinations' development. We also have to highlight that the Lake Balaton region, together with the almost complete tourism industry in Hungary, is characterized by strong spatial and temporal concentration [15]. Since, on the one hand, the majority of the tourists are visiting the capital (Budapest) and Lake Balaton, and in the latter case the time of the visit is during the summer high season, a tourism development strategy to find sustainable ways for tourism development is strongly needed in the second-most visited region of the country. The aim of this paper is to provide valid and reliable primary information based on the demand of cycling tourists around Lake Balaton.

As a mainly mass tourism-based destination, Lake Balaton already possesses a certain amount and quality of cycling tourism supply, but further developments are inevitable in order to raise its standards both from a quality and quantity perspective [16]. This is also supported by the governmental project entitled "Development of the cycling tourism services at Lake Balaton", of which the main objective is to create comprehensive cycling tourism developments activating and involving the lakeside and the background settlements. It is also an important aim that the new routes should cover itineraries where the diverse landscape features provide new visual experiences for the visitors. On the one hand, with this new approach, the cycling tourists can be acquainted with new and undiscovered areas of the destination, extending the already established (but now characterized by low-quality standards) Balaton Cycle Circuit. On the other hand, the newly involved settlements can benefit from growing visitor flow as well. This approach supports sustainability, since the spatial and temporal concentration of the tourists can be, to a certain extent, decreased in the Balaton region, and the wellbeing and life quality of the local population and the local economy can also be supported, together with the principle of the protection of the natural environment [17,18].

This tourism development approach can reposition the destination, especially in terms of active cycling tourism, providing an expected growth of visitors parallel with the decrease of seasonality. The integrated and complex development will comprehensively support the infrastructure of cycling tourism in the Balaton destination, where the primary objectives are based on the involvement of new target areas, settlements and landscape features, together with the development of creative and innovative services in order to create a more quality-oriented experience for the visitors and also for

the local population. The new developments will improve accessibility together with the decrease of spatial and temporal concentration.

The study demonstrates a basically bottom-up approach for the creation of these future developments. The authors investigated—among other market segments—the needs of the most reliable tourism target group in this case, the active cycling tourists, in order to receive primary information on their actual needs and suggestions for cycling tourism development around Lake Balaton. This bottom-up approach supports the sustainable tourism development strategy of the destination and most probably will further strengthen the enhancement of the local employment and the preservation of the already existing workplaces, and will decrease the impact of seasonality on labor force demand. All these impacts can directly and positively influence the life quality of the local population and can decrease the labor force migration from the region.

## 2. Literature Review

The world market of tourism has become a contiguous and interdependent system, in which both the demand and the supply sides have gone through considerable changes in regard to space and time, in quantitative and qualitative components alike [19,20]. Newer and newer areas are integrated into international and domestic tourism as well, and in the intensifying competition only those attractions, destinations or touristic actors can stay alive to meet the growing expectations of tourists [21].

In recent decades, one of the new centers of gravity in the economy has been the spectacular growth of the service sector within the global and national economies, including the strengthening of the role of tourism as well [22,23]. We also have to highlight, furthermore, the regional embeddedness of tourism. The presence of the locality is very important, as the overwhelming majority of the attractions of any time are built on the values of a respective place, region or natural landscape [24].

Additionally, by the construction of an adequate infra- and supra-structure for the tourism supply of a region or place, tourism also becomes the maker of territorial processes, as these processes affect, among other topics, settlement structure, employment, the development of the regional connections, the development of the environment and lifestyle and quality of life [25,26]. These socioeconomic and environmental features may also bear risks factors in themselves, which of course will clearly influence tourists in their travel decisions [27–29].

Tourism is one of the most dynamically growing industries of the 21st century and it is a phenomenon affecting a large proportion of society. Parallel with the formation of total tourism, tourism researchers also intend to gain complex information on the different factors of this complex system. Out of these research fields, the survey of consumer habits and attitudes towards tourism plays a highlighted role [30].

According to the perception of the authors, active or physical activity-based tourism is very much connected to a wide range of recent trends in tourism. From the point of view of demand, seeking experiences has become very important in recent decades [31,32]. A growing percentage of tourists intend to spend the time for recreation in a more active and responsible way, which means they need more experiences and activities. This more active lifestyle can also help to achieve the flow experience, so the involvement in different sports activities can result in a happier life as well [33,34].

Another new trend in tourism can be connected to social and demographic changes. Due to these changes, the increase in the ratio of seniors, the changes in family structure, the general increase in the education level and the increase in the need for learning and knowledge will all positively influence the number of tourists making inquiries for and taking part in active tourism [35]. Parallel with the aforementioned trends, we have to highlight one of the most recent ones as well, namely, a health-conscious lifestyle, which can be well fitted to active tourism as well [36,37].

Taking into consideration the history of activity-based or adventure tourism, it can be considered one of the new tourism products, since it became increasingly popular from the second half of the 20th century [31]. The spectacular growth of active tourism appeared parallel with a growing demand to break away from classic mass tourism and find such leisure activities as hiking, trekking, cycling and

other forms of active and health-conscious lifestyles [38,39]. Apart from the usual motivations, the main push factors are the physical and psychical impulses of the tourist, the more health conscious lifestyle and the need to spend our leisure time outdoors, in a natural environment. The basis for this tourism product is provided by the need for physical activity, closeness to nature and adventure-based experiences [40]. Of course, such activities can be practiced indoors as well [41].

Numerous general and destination-based studies prove that cycling can be classified among the most favored leisure activities, thus cycling tourism is one of the most popular active tourism products [42–45]. In the English language, this tourism product is often referred to as cycling tourism, bicycle touring or bicycle tourism. The increasing importance of cycling tourism is well-described by Lamont, who states, "The relationship between cycling and tourism is increasingly attracting scholarly attention as cycling experiences a resurgence as a recreational, leisure and sporting activity" [46] (p. 5). The growing interest in cycling tourism can be globally detected in a greater extent, mainly from the end of the 1990s, which is also supported by the increasing extent of global bicycle sales [47,48]. Ritchie and Hall determined in 1999 that the "development of infrastructure to support leisure and recreational cycling, and to stimulate tourism, is becoming increasingly prevalent as planners and policy-makers recognise the potential for cycling to contribute to economic revitalisation, particularly in rural communities" [48] (p. 6).

The definitions of bicycle or cycling tourism are also diverse and sometimes differently interpreted. The authors mostly accept the definition of Sustrans, the UK charity promoting sustainable transport, which created the National Cycle Network in the United Kingdom. They define cycle tourism as "recreational visits, either overnight or day visits away from home, which involve leisure cycling as a fundamental and significant part of the visit" [46] (p. 9).

Lamont identifies the most important aspect of these conceptual definitions: "As such, cycling, involving active or passive participation, should be the main purpose of a trip to be considered a bicycle tourism trip" [46] (p. 15). This approach is also supported by Lumsdon [49] and Ritchie [47].

Based on this research trend, academic interest started to focus on different aspects of cycling tourism based on local and regional investigations [48,50–52], on horizontal aspects such as motivation, travel and consumer behavior [53,54], on decision making processes [55–57] and on sustainable aspects as well [49,58].

The role and importance of this type of active tourism are also proved by the fact that destination-based, national and even international strategies are constantly elaborated for the understanding, development and enhancement of this sustainable form of tourism [50]. One of the best examples of the recognition of the importance of cycling tourism as a form of sustainable tourism can be connected to the European Union. The *EU Cycling Strategy. Recommendations for Delivering Green Growth and an Effective Mobility in 2030* was published in 2017, raising awareness on the importance of green growth and effective mobility systems in the European Union [59]. The other important European initiative in this respect is the Eurovelo, an initiative of the European Cyclists' Federation that aims to provide safe, high-quality cycling routes across the continent, minimizing interaction with motor vehicles (https://en.eurovelo.com/).

Among the literature dealing with customer behavior and cycling tourism, the aspects of the investigation concerning customer involvement and co-creation inspired the present research. One of the most important sources of inspiration was published by Komppula and Lassila, where the authors intended to assess the opportunities for co-creation by studying the applicability of different modes of customer involvement in tourism at different stages and/or for different purposes of service development [60]. The preceding studies of consumer or user involvement in service development are established by aspects of marketing studies [61–64]. This approach, connected with the idea of bottom-up initiatives and sustainable and responsible tourism planning, soon inspired the academics of tourism for further studies. As a result, the first comprehensive studies were published dealing in general with the opportunities of consumer involvement in tourism development [65,66].

In 2011 Hjalager and Nordin [67] published a review on the methods of user-driven innovation on tourism, and in 2010 Ritchie et al. [53] applied the concept of enduring involvement, in conjunction with tourist motivational theory, to segment and better understand cycle tourist behavior and intentions. By doing so, they suggested that the results of this very heterogenic market segment can be well utilized in product development and marketing communication. The present article strongly relies on the message and results of Ritchie et al., further adding that, based on the literature review, the research of this scope is not yet complete [53].

In accordance with the National Tourism Development Strategy 2030 of Hungary, one of the sub-products of active and nature-based tourism is cycling tourism. Based on this document, we can declare that one of the USPs (Unique Selling Propositions) of the Hungarian active tourism supply is the lakes that can be cycled around and the diverse landscape features [68]. The Government of Hungary also realized the importance of cycling tourism in terms of alternative and sustainable tourism development in the country and thus launched a general active tourism development project from EU and national sources in 2017, in order to establish and develop new infrastructural developments for active tourism and ecotourism. Emphasizing the importance of this tourism product, the National Tourism Development Strategy 2030 highlights that one of the USPs of Hungarian tourism are its cycling tourism endowments, and highlights the role of basic infrastructure development for cycling tourism networks as well [68].

## 3. Materials and Methods

During the survey, in order to reveal the demand side of active cycling tourism related to Lake Balaton, the authors used both qualitative and quantitative research methods. We applied triangulation; i.e., data were collected from different sources: qualitative surveys (focus group discussions, structured interviews) and a quantitative questionnaire (computer-assisted data collection) [69]. Regarding qualitative research, focus group discussions and structured interviews were carried out. The survey was targeted to explore the background reasons, motivations and attitudes of the tourists, in order to understand the behavior of the cycling tourist. The understanding of the ideas and attitude of the service providers and the decision-makers was also an important aspect of this research.

During the focus group researches (8–10 persons/focus group) the invited participants were interviewed, covering three major topics with the help of a moderator: the introduction of the cycling guests, cycling trends; the further development of cycling services; and the functions of a cycling center. The focus group included people with active motivations towards cycling and cycling tourism. The same questions were asked in all the 5 locations (Veszprém, Hévíz, Balatonfüred, Keszthely and Balatonföldvár) and the focus group meetings (6 altogether) lasted 1.5–2 hours.

The members of the focus groups were chosen from the same target group but with altering lifestyle and demographical features. Concerning their occupation, they could be connected to cycling tourism directly (service providers) and indirectly (decision-makers): e.g. cycling shop owner, cycling service station and bicycle rental shop owner, tour guide, tour and race organizer, settlement developer, member of the local government, tourism service provider, member of a tourism association, tourist information bureau associate, leader of a cycling association, spinning trainer, hobby cyclist.

During the research, the authors also carried out deep interviews with professionals, covering 1.5–2 hour interviews per person. Here we specifically intended to understand the opinions of people who are experts on cycling and cycling tourism (project leaders, decision-makers, entrepreneurs in cycling tourism, tour leaders, representatives of forestry workers, representatives of transport enterprises, cycling associations, professionals, representatives of other interest groups). After the first 8 interviews the other respondents were chosen by the snowball method, so the new respondents were proposed by the original interviewees. Altogether 22 people were interviewed with this method. The number of these interviews cannot be considered representative, but the sample was carefully chosen in order to represent such groups or subcultures who have determining and reliable opinions related to the reviewed topics [70].

The quantitative research was based on the descriptive statistics and the relationship analyses. The authors evaluated the answers of the questionnaire survey with descriptive statistics revealing the frequency, their distribution and, where it was possible, the mean values, the standard deviation and the mode. The answers for the open questions were classified into categories and then the frequency of these categories was created using tables and word clouds.

The strength between the answers received for the questions was investigated with relationship analyses. The results were defined on a 5% significance level. Cramer's V coefficient and Kendall's tau ($\tau$) rank correlation coefficient were applied to determine the strength of relationships. Cramer's V measures association between nominal variables and it can vary from 0 to 1. Kendall's tau measures rank correlation between ordinal variables and varies from $-1$ to 1.

## 4. Results

### 4.1. Primary Data Analysis Focusing on the Development Proposals

As a first step, we sum up the demographic characteristics of the 809-person sample in order to receive the social features of the respondents. There were 58% males and 42% females in the sample. According to their age, 48% of the respondents belonged to the 25–44-year-old age group and 43% to the 45–64 age group. The respondents provided data from 355 different settlements of Hungary.

The majority (81%) did not possess holiday houses or second homes in the Balaton region and their household typically covered 2–4 persons (78%). According to their highest education level, most of the respondents possessed a higher education (diploma) (62%) or secondary level education (36%) qualification. Based on their occupation, 37% were intellectual employees, followed by entrepreneurs (16%), middle or higher managers (16%) and physical employees (12%). The authors also intended to map their general way of feeling with two questions. One of them was how happy they feel themselves (based on a 1–10 scale, where 10 meant that they are perfectly happy) and the other question was related to their state of health, where 10 meant that they have an outstanding state of health. The greatest group valued their happiness level and state of health at eight (35% and 31%). There was no one who selected one or two.

In further investigations, we moved towards the survey of the cycling activities and habits of the sample. First, we asked about the regularity of cycling in free time. The majority (48%) indicated that they cycled on a weekly basis, but within this sample it was quite diverse whether they cycled on weekdays or weekends. The second-greatest group (23%) indicated that they cycled on a daily basis. When we asked about their objective for cycling there were two typical answers: 85% of them indicated preserving and developing sport, fitness and shape, and 79% indicated free time activities and leisure travel (several answers could be indicated).

The 809 respondents used a total of 1362 bicycles. The majority, 91%, were traditional bicycles and 9% were e-bikes. According to the type of bicycle, 26% of the used bikes were road bikes, 26% were mountain bikes, 25% were cross-trekking bikes, 16% were urban bikes and 6% were other types.

During the touring or going for a trip with bicycle the greatest respondent group (25%) liked to cover 41–60 kilometers. The second most frequent answer (20%) was more than 80 kilometers, followed by 60–80 kilometers (18%). During cycling, most of them cycled alone (30%), and 25% cycled with a maximum of four people, such as with partners, friends and relatives.

We were also inquisitive about whether the respondents ever bicycled on designated Hungarian bicycle routes in the last three years. Most of them (78%) cycled on the Balaton Cycle Circuit (around the Lake), followed by the Balaton region (78%) (several answers could be indicated). These data were very important in order to assess the respondents' opinions about the cycling opportunities around Lake Balaton. We also offered nine different bicycle tracks around Lake Balaton to indicate which of them they would like to tour on. All of these tracks were very popular among the respondents, since 65–90% of the sample indicated certain cycling routes. The least favored sections were at the southern

shore of the lake (65%) and the most favored sections were on the northern shores (90%), which might be in relation with the different relief conditions of the two sections as well.

In connection with the transport of bicycles, we asked two questions where multiple answers could be indicated. First, we asked what kind of transport modes they prefer to transport their bicycles if they could travel from their dwelling place to Lake Balaton. Two answers were chosen at a very high rate, namely train (72%) and cars (72%) followed by the ferries (50%) and ships (46%) of Lake Balaton. Only 14% of the respondents declared that they do not intend to demand any transport services.

We also asked the respondents, if they were to start a bicycle tour around Lake Balaton, with which transport mode would they combine cycling. The majority would combine their cycling activities with travelling on the train (55%), then with car (46%), ferries (46%) and ships (39%). Here 27% indicated that they would not combine cycling with any other forms or modes of transport.

A very important aspect of the present paper is that, as follows, we listed 25 types of services and infrastructural demand aspects and inquired about their importance in the selection procedure of the cycle tour or excursion. For this question, the respondents could indicate their opinion on a 1–4 scale where 1 meant "not important at all" and 4 meant "very important". The following factors were typically considered to be less important:

- Bike transfer service (between accommodation and bike tour starting point)
- Public Wi-Fi at bike stops
- Electric charging point (for e-bike)
- E-bike rental possibility
- Traditional bike rental possibility

The last two aspects were most probably indicated because we asked respondents who regularly ride a bike, so they all possessed some kind of a bicycle. Further, since only 9% of the respondents owned an e-bike, electric charging points were not really important for the majority. The demand for Wi-Fi can be reasoned by the fact that according to their social status the majority of the respondents belonged to such groups who are characterized by higher spending characteristics.

The most important aspects for the respondents were:

- Unified board and sign system along the designated path
- Clear information maps along the cycling road/at rest areas
- Drinking water along the bicycle road/path
- Possibility to buy food along the bicycle road/path
- Eating opportunities: buffet
- Safe storage for the bike at the accommodation/sites (secured bike storage)
- Public toilets along the bicycle road
- Parking place where the tour starts
- Designated asphalt bicycle path

One of the most important aspects in terms of our research was when we asked the active cyclist respondents in an open question what kind of development proposals they have in order to establish and further strengthen a bicycle route network in the Balaton region. A total of 43% of the sample gave answers for this open question, which are listed and categorized in the following figure (Figure 4).

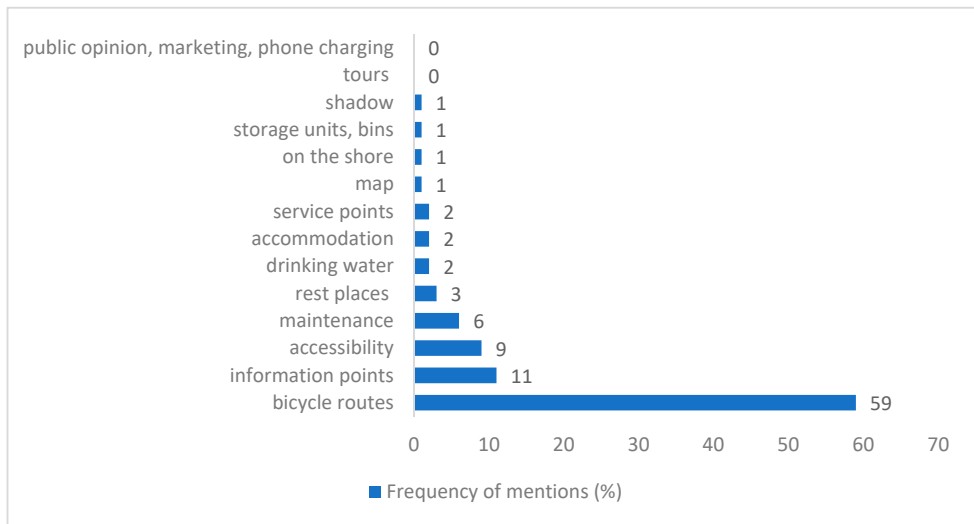

**Figure 4.** What suggestions do you have for the development of the Balaton region bicycle tour network? (the main topics are represented in the Figure and the further descriptions under the Figure). Source: the authors.

Further, the authors applied relationship analyses in order to reveal deeper knowledge about the answers of the closed questions. Table 1 shows the research model that categorizes the closed questions of the survey into two groups: potential explanatory variables (demographic indicators) and response variables (related to cycling habits.) The main aim of the research was to explore the relationships between them. The question mark in the green arrow represents that we are looking for the answer to the following research question: do demographic characteristics (D) significantly influence cycling habits (C]) and if so, how strong is that relationship? (Table 1).

**Table 1.** The scheme of the research model. Source: the authors.

| Potential Explanatory Variables Demography (D) | ? | Response Variables Cycling Habits (C) |
|---|---|---|
| (D1) Gender | | (C1) How often do you cycle in your free time? |
| (D2) Age | | (C2) What is the reason for occasional cycling? |
| (D4) Do you have a vacation home or a second home in the Balaton region (apartment, land, flat that is not permanent residence)? | | (C3) What type of bicycle do you use presently? |
| (D5) Number of household members | | (C4) What distance would you take daily during your cycling tour or trip |
| (D6) Qualification | | |
| (D7) Job | | |
| (D8) To what extent do you feel happy? (1–10 scale) | | (C5) With whom would you cycle during your tour/trip? |
| (D9) How would you judge your health status? (1–10 scale) | | (C6) In the last 3 years have you cycled in the following areas? |
| | | (C8) In what ways do you want your bicycle to be transported (from your residence/accommodation to the starting point of a Lake Balaton tour)? |
| | | (C9) If you take a cycling tour by Lake Balaton what transportation would you combine this tour with? |
| | | (C10) How important are the following services/infrastructures when choosing a bike tour or trip destination? (1–4 scale) |
| | | (C11) Why would you recommend cycling to others? |

Color Key: by the measurement scale of the question:
- Nominal.
- Ordinal

In the research model the background colors refer to the measurement scales the answers for the certain questions can be measured with. As a final result, the strength of 5050 possible relationships were examined between the answers. There were relatively few (and quite weak) significant relationships between the demographic variables and the cycling habits. Within them, one of the strongest was between gender and the distance the respondents intended to take during a cycling trip. Based on these correlations we could determine the following (basically expected) "the more/less ... the more/less" kind of statements (Table 2).

**Table 2.** The detected "the more/less ... the more/less" statements of the survey.

| | |
|---|---|
| The older (younger) someone is, the less important (the more important) to her/him is the | • traditional bike rental possibility.<br>• e-bike rental possibility.<br>• bike transfer service (between accommodation and bike tour starting point).<br>• eating possibility (restaurant).<br>• possibility to buy food along the bicycle road/path.<br>• drinking water along the bicycle road/path.<br>• active tourism attraction (adventure park, bobsleigh track) along the bike road/path. |
| The more (the fewer) people are in a household, | • the shorter (the longer) distance they would take daily during a cycling tour on the trip.<br>• the more (the less) important to her/him is the designated asphalt bicycle path.<br>• the more (the less) important to her/him is the parking place where the tour starts. |
| The higher (the lower) someone's qualification, | • the less (the more) regularly they cycle in their free time.<br>• the less (the more) important to her/him is the<br>• electric charging point (for e-bike).<br>• the less (the more) important to her/him is the e-bike rental possibility.<br>• the less (the more) important to her/him is the active tourism attraction (adventure park, bobsleigh track) along the bike road/path. |
| The happier (the less happy) someone feels, the more (the less) important is to them the | • public toilet along the bicycle road. |
| The healthier (the less healthy) someone feels, | • the more (the less) regularly they cycle in their free time.<br>• the longer (the shorter) distance they would take daily during a cycling tour on a trip. |

Further, we demonstrate those results which are connected to the most important response variables (C10). The original question was, "How important are the following services/infrastructures when choosing a bike tour or trip destination?". The answers were received on a 1–4 scale. Tables 3 and 4 summarize the results of the relationship analyses between this (C10) and demography (D). Table 2 demonstrates the significant results of the Cramer indicators in order to detect the relationship strength between the answers on a nominal scale (Table 3). It shows that the importance of the listed 25 services/infrastructures is most affected by gender (in 14 cases out of 25). In comparison, the effect of having a second home or job is negligible (fewer and weaker relationships).

**Table 3.** The results (Cramer's V) of the relationship analyses demography (D) and cycling habits (C). Source: the authors.

| | | | | Potential Explanatory Variables: Demography [D] | | |
| | | | | D1 Gender | D4 Second Home | D7 Job |
|---|---|---|---|---|---|---|
| Response variable: Cycling habits [C] | [C10] How important are the following services/infrastructures when choosing a bike tour or trip destination? (1-4 scale) | 1. | designated asphalt bicycle path | 0.124 | - | - |
| | | 2. | designated forest route | 0.128 | - | - |
| | | 3. | designated agricultural (no asphalt) route | - | - | - |
| | | 4. | a smaller level difference on the itinerary (ridges, slopes) | 0.106 | - | - |
| | | 5. | bike service point/centre (technical assistance) | 0.188 | - | - |
| | | 6. | shelters and rest points (in case of rain) | 0.148 | - | - |
| | | 7. | parking place where the tour starts | - | 0.017 | 0.137 |
| | | 8. | public toilet along the bicycle road | 0.168 | - | - |
| | | 9. | electric charging point (for e-bike) | - | - | - |
| | | 10. | public Wi-Fi at bike stops | 0.147 | - | - |
| | | 11. | traditional bike rental possibility | 0.118 | - | - |
| | | 12. | e-bike rental possibility | - | - | - |
| | | 13. | bike transfer service (between accommodation and bike tour starting point) | 0.125 | 0.091 | - |
| | | 14. | safe storage for the bike at the accommodation/sites (secured bike storage) | - | - | - |
| | | 15. | type-specific bicycle stands at rest areas and service providers (restaurant, buffet) | 0.137 | - | - |
| | | 16. | eating possibility: buffet | - | - | - |
| | | 17. | eating possibility: restaurant | - | - | - |
| | | 18. | possibility to buy food along the bicycle road/path | - | - | - |
| | | 19. | drinking water along the bicycle road/path | - | - | - |
| | | 20. | clear information maps along the cycling road/at rest areas | 0.142 | - | - |
| | | 21. | unified board and sign system along the designated path | - | - | - |
| | | 22. | mobile application (in English) of the designated path | 0.102 | - | - |
| | | 23. | lookout point, natural sights, short study path along the road (for free) | 0.142 | - | - |
| | | 24. | cultural attractions (fortress, castle, museum, gardens) along the bike road/path | - | - | - |
| | | 25. | active tourism attraction (adventure park, bobsleigh track) along the bike road/path | 0.103 | - | - |

Notes: non-significant results. Color Key: by the strength of the significant results.

weaker ▭ stronger.

**Table 4.** Relationship analysis matrix, τ indicator between demography [D] and cycling habits [C]. Source: the authors.

| | | Code 1 Means: The | Potential Explanatory Variables: Demography [D] | | | | |
| | | | D2 Age | D5 Household | D6 Qualification | D8 Happiness | D9 Health |
| | | | Youngest | Least | Lowest | Least Happy | Least Healthy |
|---|---|---|---|---|---|---|---|
| C | How important are the following services/infrastructures when choosing a bike tour on (1–4) | 1. | - | 0.065 | - | - | - |
| | | 2–6. | - | - | - | - | - |
| | | 7. | - | 0.091 | - | - | - |
| | | 8. | - | - | - | 0.059 | - |
| | | 9. | - | - | −0.091 | - | - |
| | | 10. | - | - | - | - | - |
| | | 11. | −0.104 | - | - | - | - |
| | least important | 12. | −0.065 | - | −0.074 | - | - |
| | | 13. | −0.077 | - | - | - | - |
| | | 14–16. | - | - | - | - | - |
| | | 17. | −0.095 | - | - | - | - |
| | | 18. | −0.083 | - | - | - | - |
| | | 19. | −0.100 | - | - | - | - |
| | | 20–24. | - | - | - | - | - |
| | | 25. | −0.139 | - | −0.064 | - | - |

Notes: non-significant results. Color Key: by the strength of the significant results

weaker ▭ stronger.

Table 4 demonstrates the significant results of the Kendall τ indicators in order to detect the relationship strength between the answers on a sequence scale (Table 3). Clearly, age has the greatest impact on the importance of the listed 25 services/infrastructures among the demographic characteristics examined.

*4.2. The Evaluation of Cycling Tourism Development from the Point of View of the Structured Deep Interviews and the Focus Group Surveys*

Based on the structured deep interviews and the focus group surveys, four major trends or directions can be determined (Figure 5). Related to the cycling route developments, the most important element is the route itself. Based on the results the most important priority is to build "a high-quality road in good environmental conditions", followed by the professional designation of the routes on which one can use the bicycle.

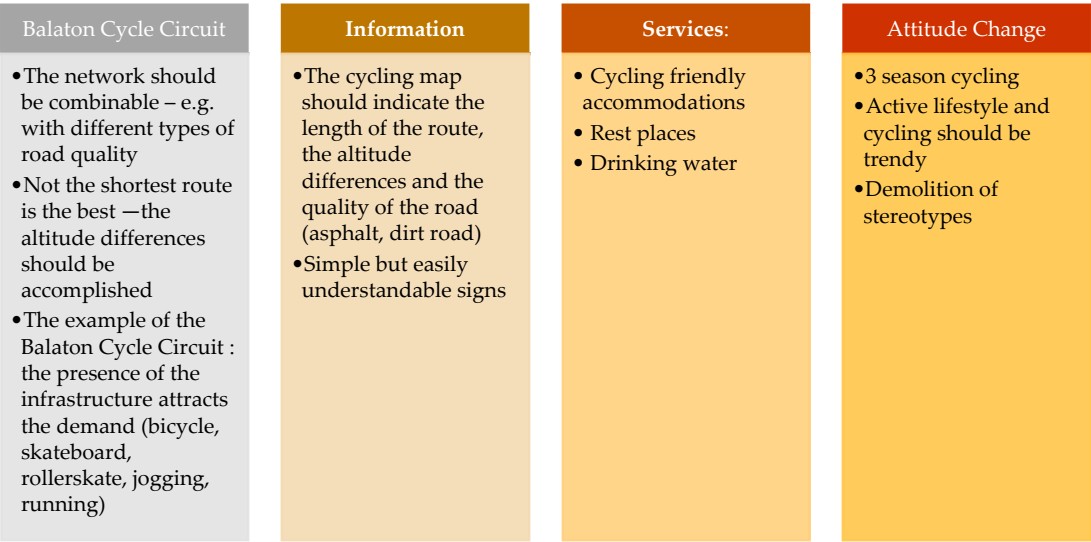

**Figure 5.** The 4 major trends and directions provided by the structured deep interviews and the focus group surveys. Source: the authors.

Based on the received answers, there is no sense to build a well-developed surfaced road below the length of 30 kilometers since this is such a length that can be desirable for the possible consumers. The surveyed responders emphasized that on the existing forest or agricultural routes the shorter (20–25 kilometers) distances are ideal, where even the hobby cyclists dare to roam. According to the professionals, most of all, the lack of information should be improved, difficulty levels should be identified and the level of safety should be emphasized. On the information maps, the cyclists intend to see routes, crossroads, altitude differences and road quality information. Taking into consideration cyclist services, the availability of drinking water, information access on maps and safe bicycle storage is of great priority.

In the case of individually organized bicycle tours, it is very important to simplify the opportunities for online planning and booking, and to further develop the creation and communication of cycling packages and convenience services.

There is a growing tendency of positive experiences with e-bikes. The respondents consider this type of cycling as a positive solution primarily for the elderly and for those with weaker physiques. With this type of cycling, "cycling is not an obstacle" and the "couple or group can stay together during the tour". Finally, in order to increase demand, all the respondent groups mentioned the need for a generation change in order to involve the younger generation in the cycling activities.

## 5. Discussion

The respondents of the research highlighted all the accentuated landscapes (Balaton Highlands, Keszthelyi Mountains, Kis (Small) Balaton, Káli Basin, Tihanyi Peninsula) and settlements (Fonyód, Badacsony, Keszthely, Szigliget and Balatonfüred) of the Balaton Region as areas that are attractive for cycling tourism. From the point of view of cycling as a mode of transport, the region is not as crowded as in a busy city and moving off from the coastline one can find multiple sub-routes as well. From the physical features, the aesthetic value of the panorama of Lake Balaton should also be emphasized. From the point of view of route development, it is also important to highlight that the settlements are quite close to each other (2–4 kilometers on the northern shore). The smaller settlements are connected with lower category roads with low traffic and are provided with guest catering places.

Our research demonstrated that there is a definite demand for leisure time cycling in the Lake Balaton region, as an activity and tourism product which highly promotes sustainability, sustainable tourism development, together with an active and trendy lifestyle. Besides cycling, the auxiliary tourism products are bathing in the summertime, religious tourism, ecotourism, wine and gastronomy, which can be connected with cycling out of the high season as well. This is an especially important feature, since the level of seasonality is high in the region. In terms of cycling and cycling tourism, the existing Balaton Cycle Circuit is definitely a strength at the Balaton Region, however, its present quality is poor.

The most critical element of cycling tourism at Lake Balaton is the route itself, the length and the quality. The existing weaknesses are connected to the experienced bad quality of the route, the missing sections, the traffic morale and the lack of service infrastructure. Many respondents mentioned that the existing bicycle route is not convenient and safe and there is no responsible body for the maintenance and renewal of the route and its related infrastructure. In the background areas, especially at the northern shore, there are routes with higher altitude differences where the physical preparedness is not enough for many, and in the region at the southwest part of the Lake (Kis Balaton) the opportunities for touring are limited. Many respondents missed the presence of cycling associations and events and emphasized that today cycling is restricted to the Balaton Cycle Circuit around the lake.

According to the respondents, the best practices for cycling tourism and cycling tourism development are in Austria, Slovenia and Italy, and so they serve as a comparison for the opportunities at Lake Balaton.

Summing up, the detected consumer preferences and development proposals of active cycling tourists would all serve for better and high-quality cycling tourism in the region. The development of cycling tourism and the mentioned preferences would serve as a basis for a slight focus change of the tourism industry at Lake Balaton from mass tourism to sustainable or to more sustainable tourism. The proposed developments would also serve as tools for economic and social development for the local population and within those settlements situated in the background areas of the lake. In case of proper achievement of the mentioned development goals, cycling tourism could serve as a win-win situation at the Balaton region for active cycling tourists, the local population and the local economy.

## 6. Conclusions

The authors believe that one of the most important results of the presented research relies on their applicability for decision-makers in terms of developing new and existing bike paths and also on creating customer opinion-based development directions. This latter approach is not widely accepted in Hungarian regional development planning practice, so the authors believe that the presented research can contribute to the strengthening of bottom-up tourism planning. The results show that the content analysis of the primary research provides well determined directions for the sustainable tourism development of cycling tourism at Lake Balaton, so customer involvement seems to be a win-win situation both for the customers (tourists), the local population and the decision-makers.

The presented research revealed those segmentation peculiarities which are needed for sustainability-oriented service developments around a mostly mass tourism-based destination. Due

to these new directions of tourism planning, the new bicycle paths and the connected services can be adequately determined, promoting the decrease of spatial and temporal seasonality in the destination. The designated route and the related development proposals, which were also channeled to the decision-making actors, greatly rely on the results of the consumer-based needs of the active cycling tourists.

As for the limitations of the research, the authors understand that the survey was not completely representative, despite the fact that a remarkable amount of multitude was involved. The other limitation of the research is that the questionnaire was anonymous, so the authors cannot expand it to panel data analysis later. In this case, the same sample cannot be interviewed again, in order to receive an evaluation to find out how effective the improvements were for the users. As for the future research directions, there is still more opportunity to widen this research to the already existing, international bike paths of Hungary (EuroVelo) in order to receive not just a regional, but a national scope of further development proposals, based on consumer involvement on the side of active cycling tourists.

**Author Contributions:** Conceptualization, K.L., Z.B. and J.C.; methodology, K.L., Z.B. and J.C.; software, Z.B.; validation, Z.B.; formal analysis, K.L., Z.B. and J.C.; investigation, K.L. and Z.B.; resources, K.L. and Z.B.; data curation, K.L. and Z.B.; writing—original draft preparation, J.C., Z.B. and K.L.; writing—review and editing, J.C., Z.B. and K.L.; visualization, J.C., Z.B. and K.L.; supervision, J.C. All authors have read and agreed to the published version of the manuscript.

**Funding:** We acknowledge the financial support of Széchenyi 2020 under the EFOP-3.6.1-16-2016-00015.

**Conflicts of Interest:** The authors declare no conflict of interest.

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
