# Peer review of "Customer Involvement in Sustainable Tourism Planning at Lake Balaton, Hungary—Analysis of the Consumer Preferences of the Active Cycling Tourists"

_sustainability, doi:10.3390/su12125174_

Round 1

Reviewer 1 Report

  • Abstract - better explanation of research design and methodology 
  • Well written paper with qualitative and quantitative research 
  • Missing section - conclusive remarks 

Author Response

Authors’ Response to the Comments of Reviewers

considering manuscript originally entitled

"Sustainable aspects of tourism development at Lake Balaton, Hungary"

submitted to Sustainability

(Tourism and Sustainability: Combining Tourist’s Needs with Destinations’ Development)

Dear Editor, Dear Reviewers,

We would like to thank you for the opportunity to re-submit our manuscript.

The following figure summarizes the topics mentioned by the Reviewers, from which the structure of the manuscript is the main shortcoming of our article. All these topics have been improved:

Reviewer

1

2

3

4

Methodology (why 22 interviews are an ideal number?)

X

Map: should be clarified

X

Structure (missing or weak sections)

Abstract

X

Introduction

X

X

Literature review

X

X

X

Discussion

X

X

Conclusion

X

X

X

X

Limitations

X

Future research

X

X

We would like to also thank you for your thorough, constructive comments. Based on Reviewers’ remarks and suggestions, the manuscript was revised. The corrected paper is attached:

  • with track changes: manuscript_corr_with track changes.docx
  • without track changes: manuscript_corr_without track changes.docx

We have made a substantial/major revision to our manuscript. Due to the major revision (and the significant changes), we could not answer for the reviewers by providing the exactly changed lines as well, but all the changes can be detected in the track changes version of the article.

We hope after the revision, the re-submitted paper meets your expectations.

Sincerely,

The authors

June 12, 2020

Reviewer 1:

Abstract - better explanation of research design and methodology 

  • The authors extended the abstract with better explanation of research design and methodology:

Abstract: The study intends to deal with an innovative tourism product development approach based on co-creation or customer involvement, related to Lake Balaton, a mass tourism-based destination of Hungary, from the point of view of the market segment of the active cycling tourists. The investigations on the opportunities of the development of cycling tourism first of all rely on the new approach of attraction and product development around the destination where it is a highly important aspect to take into consideration the consumer preferences of the most important related group of tourists, the active cycling tourists. The sustainable approach of tourism product development provides also an opportunity to decrease the spatial and timely concentration of tourism, which is largely on the summertime season. The aim of this study is to explore the aspects of the customers’ demand for tourism development in terms of cycling tourism with the help of primary data collection in order to receive results that would provide adequate directions for sustainable tourism development in the destination. Revealing the demand side of the active cycling tourists related to Lake Balaton, the authors used both qualitative (focus group discussions and structured interviews) and quantitative questionnaire survey (computer-assisted data collection) research methods. The latter online surveys were carried out in November and December 2019 and resulted with an appraisable sample of 809 questionnaires. As for the method, descriptive statistics and relationship analyses were applied. More than five thousand (5050 pieces) possible relationships was examined between the closed answers of the questionnaire by Kendall's rank correlation coefficient (τ) and Cramer's V, depending on whether they can be measured on a nominal or ordinal scale. The results show that the content analysis of the primary research provides well determining directions for the sustainable tourism development of cycling tourism at Lake Balaton, so customer involvement seems to be a win-win situation both for the customers (tourists) and the decision makers.

Well written paper with qualitative and quantitative research 

  • Thank you for this comment

Missing section - conclusive remarks 

  • The authors understood and completely agreed with this opinion, so we have made significant changes in the Conclusions and Discussion chapters. We made efforts on providing a more concise description focusing on the most important results, their importance for the academic sphere, the limitations of the research, and the future research design as well.
  • Thank you very much for your opinion and efforts in reviewing our article.

The Authors

Reviewer 2 Report

Overall, the present manuscript is related to sustainability development in a mass tourism destination. Therefore, it would fit under the journal's areas of study. However, I feel it presents serious flaws in structure and content that might be difficult to easily be amended in a major revision. 

The manuscript presents an original research but the work does not report scientifically sound experiments nor provides a substantial contribution to knowledge and academia. From my point of view, the manuscript (and research behind it) seems more like a presentation of results from a consultancy project than a proper academic publication. The value and contribution for the tourism development policy and strategies in the specific area of study is undeniable, though.

The manuscript lacks a strong introduction in which the purpose of the work and its significance is clearly defined, as well as a careful revision of the current state of research and key publications (although an attempt has been made at the beginning of the results section). Actually, it remains unclear what is exactly the core theoretical background for the study: is it cycling/active tourism, mass tourism, sustainable consumer behaviour? Although the aim of the paper would be more experimental than theoretical, I think that a solid academic contribution has to present some sort of solid background so the discussion can be rich and strong (which is not in its present form).

The Materials and Methods are well described but the Results section does not provide a concise description of the experimental results nor a profound interpretation of the conclusions that can be drawn. This part would require a huge effort of synthesis aimed to summarize the most important results (descriptive part) and to focus on interpreting them in light of the theoretical background taken as a reference. Implications of the results obtained are quite clear, although these are certainly focused on the specific area of study and would be convenient to try to emphasize to what extent these would be applicable to other similar cases.

Author Response

Authors’ Response to the Comments of Reviewers

considering manuscript originally entitled

"Sustainable aspects of tourism development at Lake Balaton, Hungary"

submitted to Sustainability

(Tourism and Sustainability: Combining Tourist’s Needs with Destinations’ Development)

Dear Editor, Dear Reviewers,

We would like to thank you for the opportunity to re-submit our manuscript.

The following figure summarizes the topics mentioned by the Reviewers, from which the structure of the manuscript is the main shortcoming of our article. All these topics have been improved:

Reviewer

1

2

3

4

Methodology (why 22 interviews are an ideal number?)

X

Map: should be clarified

X

Structure (missing or weak sections)

Abstract

X

Introduction

X

X

Literature review

X

X

X

Discussion

X

X

Conclusion

X

X

X

X

Limitations

X

Future research

X

X

We would like to also thank you for your thorough, constructive comments. Based on Reviewers’ remarks and suggestions, the manuscript was revised. The corrected paper is attached:

  • with track changes: manuscript_corr_with track changes.docx
  • without track changes: manuscript_corr_without track changes.docx

We have made a substantial/major revision to our manuscript. Due to the major revision (and the significant changes), we could not answer for the reviewers by providing the exactly changed lines as well, but all the changes can be detected in the track changes version of the article.

We hope after the revision, the re-submitted paper meets your expectations.

Sincerely,

The authors

June 12, 2020

Reviewer 2:

Overall, the present manuscript is related to sustainability development in a mass tourism destination. Therefore, it would fit under the journal's areas of study. However, I feel it presents serious flaws in structure and content that might be difficult to easily be amended in a major revision. 

  • The authors completely agreed with this opinion and so made significant changes in the complete article in terms of improving its structure: a new and more detailed literature review chapter was added, and we made efforts on providing a more concise description focusing on the most important results, their importance for the academic sphere, the limitations of the research, and the future research design as well.

The manuscript presents an original research but the work does not report scientifically sound experiments nor provides a substantial contribution to knowledge and academia. From my point of view, the manuscript (and research behind it) seems more like a presentation of results from a consultancy project than a proper academic publication. The value and contribution for the tourism development policy and strategies in the specific area of study is undeniable, though.

The manuscript lacks a strong introduction in which the purpose of the work and its significance is clearly defined, as well as a careful revision of the current state of research and key publications (although an attempt has been made at the beginning of the results section). Actually, it remains unclear what is exactly the core theoretical background for the study: is it cycling/active tourism, mass tourism, sustainable consumer behaviour? Although the aim of the paper would be more experimental than theoretical, I think that a solid academic contribution has to present some sort of solid background so the discussion can be rich and strong (which is not in its present form).

  • The authors strengthened the abstract and the introduction chapter as well, with which we managed to define more clearly the purpose, objectives and scientific significance of the article. As mentioned above a separate and more detailed literature chapter was also added focusing on the relevant literature providing the theoretical background for the article mainly based on consumer involvement and consumer behavior.

The Materials and Methods are well described but the Results section does not provide a concise description of the experimental results nor a profound interpretation of the conclusions that can be drawn. This part would require a huge effort of synthesis aimed to summarize the most important results (descriptive part) and to focus on interpreting them in light of the theoretical background taken as a reference. Implications of the results obtained are quite clear, although these are certainly focused on the specific area of study and would be convenient to try to emphasize to what extent these would be applicable to other similar cases.

  • The authors understood and completely agreed with this opinion, so we have made significant changes in the Conclusions and Discussion chapters. We made efforts on providing a more concise description focusing on the most important results, their importance for the academic sphere, the limitations of the research, and the future research design as well.

  • Thank you very much for your opinion and efforts in reviewing our article.

The Authors

Reviewer 3 Report

The paper consists of a quite interesting research work with local scale interest. The research gives important outcome for Hungary and cycle tourism. 
However, according to my opinion, some changes could be implemented in order for the paper to be improved. More specific:

Lines 40-45: Authors could mention why this research is important

Line 88: A short chapter should be added related to literature review concerning customer behavior and preferences or even case studies on that topic. This brief review could also be a small part of the introduction or a seperate unit.

Chapter 2: Authors should document the methodology used, i.e. why questionnaire survey and interviews (probably for triangulate data),why 22 interviews are an ideal number?

Line 226: The map could be clarified, i.e.  for areas outside the study area, a grey scale or transparency filter could be used, the legend could be presented on a white font in order to be easier readable. 

Chapter 4: Are there any general conclusions dirived by this specific research? A brief reference to that and future research.

I think that this paper is the result of a serious research and I recommend it to be published, after minor revision.

Author Response

Authors’ Response to the Comments of Reviewers

considering manuscript originally entitled

"Sustainable aspects of tourism development at Lake Balaton, Hungary"

submitted to Sustainability

(Tourism and Sustainability: Combining Tourist’s Needs with Destinations’ Development)

Dear Editor, Dear Reviewers,

We would like to thank you for the opportunity to re-submit our manuscript.

The following figure summarizes the topics mentioned by the Reviewers, from which the structure of the manuscript is the main shortcoming of our article. All these topics have been improved:

Reviewer

1

2

3

4

Methodology (why 22 interviews are an ideal number?)

X

Map: should be clarified

X

Structure (missing or weak sections)

Abstract

X

Introduction

X

X

Literature review

X

X

X

Discussion

X

X

Conclusion

X

X

X

X

Limitations

X

Future research

X

X

We would like to also thank you for your thorough, constructive comments. Based on Reviewers’ remarks and suggestions, the manuscript was revised. The corrected paper is attached:

  • with track changes: manuscript_corr_with track changes.docx
  • without track changes: manuscript_corr_without track changes.docx

We have made a substantial/major revision to our manuscript. Due to the major revision (and the significant changes), we could not answer for the reviewers by providing the exactly changed lines as well, but all the changes can be detected in the track changes version of the article.

We hope after the revision, the re-submitted paper meets your expectations.

Sincerely,

The authors

June 12, 2020

Reviewer 3:

The paper consists of a quite interesting research work with local scale interest. The research gives important outcome for Hungary and cycle tourism. 
However, according to my opinion, some changes could be implemented in order for the paper to be improved. More specific:

Lines 40-45: Authors could mention why this research is important

  • Thank you for this comment, we made changes in the article accordingly highlighting the importance of the research

Line 88: A short chapter should be added related to literature review concerning customer behavior and preferences or even case studies on that topic. This brief review could also be a small part of the introduction or a seperate unit.

  • Thank you for this comment: a separate and more detailed literature chapter was also added focusing on the relevant literature providing the theoretical background for the article mainly based on consumer involvement and consumer behaviour.

Chapter 2: Authors should document the methodology used, i.e. why questionnaire survey and interviews (probably for triangulate data),why 22 interviews are an ideal number?

  • In the methodology chapter, we provided a more detailed description why the applied methodology was used.

Line 226: The map could be clarified, i.e.  for areas outside the study area, a grey scale or transparency filter could be used, the legend could be presented on a white font in order to be easier readable. 

  • Thank you, we corrected the map accordingly.

Chapter 4: Are there any general conclusions dirived by this specific research? A brief reference to that and future research.

  • The authors understood and completely agreed with this opinion, so we have made significant changes in the Conclusions and Discussion chapters. We made efforts on providing a more concise description focusing on the most important results, their importance for the academic sphere, the limitations of the research, and the future research design as well.

I think that this paper is the result of a serious research and I recommend it to be published, after minor revision.

  • Thank you very much for your opinion and efforts in reviewing our article.

The Authors

Reviewer 4 Report

Thanks to the authors for taking up the topic of cycling tourism in their article.
I think that article is analytically correct but requires many substantive corrections. Particularly significant shortcoming of this article is its structure.

The article submitted for review does not contain a chapter on literature review. Although the authors review a small resource of scientific papers but unfortunately put it in the results.  Perhaps this is a technical error because in the third chapter there are two subsections with number 3.1. In turn, in the Discussion chapter, a rather summary of the research is included. Unfortunately, this is incomprehensible.

It is necessary to develop a scientific discussion on the relationship between cited scientific publications and the results of own research. A chapter on literature review on the topic should also be separated. The scope of cited publications also needs to be supplemented as it is too small.

The article should be supplemented with restrictions resulting from the conducted research and also indicate future directions of research on the topic.

The improvement of the article should also take into account the comprehensive answer to the question - how can the obtained research results in one area (region) affect the organization of tourism in other regions?

I wish the author good luck in improving their article.

Author Response

Authors’ Response to the Comments of Reviewers

considering manuscript originally entitled

"Sustainable aspects of tourism development at Lake Balaton, Hungary"

submitted to Sustainability

(Tourism and Sustainability: Combining Tourist’s Needs with Destinations’ Development)

Dear Editor, Dear Reviewers,

We would like to thank you for the opportunity to re-submit our manuscript.

The following figure summarizes the topics mentioned by the Reviewers, from which the structure of the manuscript is the main shortcoming of our article. All these topics have been improved:

Reviewer

1

2

3

4

Methodology (why 22 interviews are an ideal number?)

X

Map: should be clarified

X

Structure (missing or weak sections)

Abstract

X

Introduction

X

X

Literature review

X

X

X

Discussion

X

X

Conclusion

X

X

X

X

Limitations

X

Future research

X

X

We would like to also thank you for your thorough, constructive comments. Based on Reviewers’ remarks and suggestions, the manuscript was revised. The corrected paper is attached:

  • with track changes: manuscript_corr_with track changes.docx
  • without track changes: manuscript_corr_without track changes.docx

We have made a substantial/major revision to our manuscript. Due to the major revision (and the significant changes), we could not answer for the reviewers by providing the exactly changed lines as well, but all the changes can be detected in the track changes version of the article.

We hope after the revision, the re-submitted paper meets your expectations.

Sincerely,

The authors

June 12, 2020

Reviewer 4:

Thanks to the authors for taking up the topic of cycling tourism in their article. 
I think that article is analytically correct but requires many substantive corrections. Particularly significant shortcoming of this article is its structure.

The article submitted for review does not contain a chapter on literature review. Although the authors review a small resource of scientific papers but unfortunately put it in the results.  Perhaps this is a technical error because in the third chapter there are two subsections with number 3.1. In turn, in the Discussion chapter, a rather summary of the research is included. Unfortunately, this is incomprehensible.

  • The authors completely agreed with this opinion and so made significant changes in the complete article in terms of improving its structure: a new and more detailed literature review chapter was added, and we made efforts on providing a more concise description focusing on the most important results, their importance for the academic sphere, the limitations of the research, and the future research design as well.

It is necessary to develop a scientific discussion on the relationship between cited scientific publications and the results of own research. A chapter on literature review on the topic should also be separated. The scope of cited publications also needs to be supplemented as it is too small.

  • Thank you, of course we accepted this opinion as well: As mentioned above a separate and more detailed literature chapter was also added focusing on the relevant literature providing the theoretical background for the article mainly based on consumer involvement and consumer behaviour.

The article should be supplemented with restrictions resulting from the conducted research and also indicate future directions of research on the topic.

The improvement of the article should also take into account the comprehensive answer to the question - how can the obtained research results in one area (region) affect the organization of tourism in other regions?

  • The authors understood and completely agreed with this opinion, so we have made significant changes in the Conclusions and Discussion chapters. We made efforts on providing a more concise description focusing on the most important results, their importance for the academic sphere, the limitations of the research, and the future research design as well.

I wish the author good luck in improving their article.

  • Thank you very much for your opinion and efforts in reviewing our article.

The Authors

Round 2

Reviewer 2 Report

The authors made a great effort in adressing most of the flaws identified in my previous report. The theoretical background, objectives, contribution, discussion of results and overall structure is now very much improved. Good job!

Reviewer 4 Report

Thank you for improving the article.  I think that it is well improved and suitable for publication.  I am asking you only to update the numbering of the bibliography.  Congratulations to the authors of the new article and I send my best wishes.